# Uncertainty Quantification of Compressor Map Using the Monte Carlo Approach Accelerated by an Adjoint-Based Nonlinear Method

**Shenren Xu, Qian Zhang, Dingxi Wang and Xiuquan Huang ***

School of Power and Energy, Northwestern Polytechnical University , Xi'an 710129, China
* Correspondence: xiuquan_huang@nwpu.edu.cn

**Abstract:** Precise and inexpensive uncertainty quantification (UQ) is crucial for robust optimization of compressor blades and to control manufacturing tolerances. This study looks into the suitability of MC−adj−nonlinear, a nonlinear adjoint-based approach, to precisely and rapidly assess the performance discrepancies of a transonic compressor blade section, arising from geometric alterations, and building upon previous research. In order to assess the practicality and illustrate the benefits of the adjoint-based nonlinear approach, its proficiency and precision are gauged against two other methodologies, the adjoint-based linear approach (MC−adj−linear) and the high-fidelity nonlinear Computational Fluid Dynamics (MC−CFD) method. The MC−adj−nonlinear methodology exhibits impressive generalization capabilities. The MC−adj−nonlinear method offers a great balance between precision and time efficiency, since it is more precise than the MC−adj−linear method in both design and near-stall conditions, yet requires approximately a thirtieth of the time of the MC−CFD method. Finally, the MC−adj−nonlinear method was utilized to conduct fast UQ analyses of the section at four distinct speeds to quantify the performance uncertainty for the compressor map. It is found that aerodynamic performance is more sensitive to geometric deviations at high speeds than at low speeds. The impact of the geometric deviations is generally detrimental to the mean efficiency.

**Keywords:** manufacturing variability; uncertainty quantification; Monte Carlo; adjoint method; compressor map

## 1. Introduction

In reality, it is impossible to avoid discrepancies between the actual shape of manufactured blades and their intended design, resulting in performance deviations from their design intent [1]. The exact performance of the blade cannot be determined until after it is manufactured [2]. The performance's mean and standard deviations [3,4] can be used to measure the level of uncertainty [5]. Robust optimization [6–8] and/or production tolerance tailoring [9] are required to reduce variability in performance. Either approach requires UQ to obtain performance statistics precisely and quickly.

The most common UQ approach is the Monte Carlo method [10] based on a Computational Fluid Dynamics (CFD) solver (MC−CFD) [11]. However, MC−CFD can be prohibitively expensive for a three-dimensional compressor, as the number of CFD simulations required for a reliable UQ is at least in the order of thousands. Considering the computational cost, surrogate models [12] and polynomial chaos expansion (PCE) [13,14] was proposed to replace the MC−CFD method. Unfortunately, it is still expensive to construct such models for a three-dimensional compressor. It is promising to construct linear and quadratic models, based on first- and second-order sensitivities. The finite difference method (FDM) and the adjoint method [15] can both be used to calculate the derivatives. Due to its high efficiency in calculating sensitivity in cases where there are several interested performance functionals, but many geometric variables, the adjoint method has attracted a

lot of attention. Marta, Rodrigues et.al. assessed the sensitivity of performance to boundary conditions by employing the adjoint method [16] and developed the adjoint solver for multi-row turbomachinery [17]. The study of the geometric sensitivity of multi-row turbomachinery [18], which catalyzed the use of the adjoint method for optimization, was highly beneficial. In terms of application for UQ, using the adjoint method to calculate the sensitivity to construct the MC−adj−linear method [19] is computationally efficient, but neglects the nonlinearity of the performance functional. The quadratic model [20] can capture part of the nonlinearity but the computational cost increases significantly with increase of the dimension. The adjoint-based nonlinear method [21] (MC−adj−nonlinear) was proposed nearly two decades ago. However, the method did not attract due attention, and no ensuing application of the method was reported in the open literature, until it was rediscovered recently [22,23]. In ref. [22,23], it was found that the MC−adj−nonlinear approach could capture some of the nonlinear relationship between performance functional alterations and geometric variations, while maintaining the same order magnitude time cost as that of the adjoint-based linear approach.

Within this context, UQ is conducted for the same case as in the Ref. [22,23] to investigate its applicability at different conditions to rule out the contingency and to demonstrate the advantages of the MC−adj−nonlinear method. Subsequently, the MC−adj−nonlinear method is applied to carry out fast UQ of the blade section's aerodynamic performance, due to manufacturing uncertainties at 50%, 80%, 100%, and 120% of the nominal speeds, to examine how geometric deviations affect aerodynamic performance at various shaft speeds.

## 2. The Adjoint-Based Nonlinear/Linear Method (Adj-Nonlinear/Linear)

This section introduces the MC−adj−nonlinear method [21,22] and the MC−adj−linear method [19]. The performance functional $J$ represents mass flow rate ($\dot{m}$), pressure ratio ($\pi$) or efficiency($\eta$), and is a function of $\boldsymbol{U}$, and $\boldsymbol{\alpha} = (\alpha_1, \alpha_2 \ldots \ldots, \alpha_M)^T$:

$$J = J(\boldsymbol{U}, \boldsymbol{\alpha}). \tag{1}$$

where, $\boldsymbol{U}$ represents the flow solution, a 6×N vector for an unstructured Reynolds-averaged Navier—Stokes (RANS) flow solver, whose turbulence equation is the Spalart—Allmaras turbulence equation. The number '6' presents the primal variables and 'N' is the nodes of the mesh. $\boldsymbol{\alpha}$ is the geometric variable vector, which is used to described the geometry of the blade. It can be the vector of coordinate points at the blade surface or the vector of design parameters like chord, thickness and so on. $\boldsymbol{U}$ and $\boldsymbol{\alpha}$ always satisfy the following symbolic flow governing equation:

$$\boldsymbol{R}(\boldsymbol{U}, \boldsymbol{\alpha}) = 0, \tag{2}$$

where, $\boldsymbol{R}$ represents the nonlinear residual. The adjoint method can be used to calculate the sensitivity of the performance functional to the geometric variable vector as follows:

$$\frac{\mathrm{d}J}{\mathrm{d}\boldsymbol{\alpha}} = \frac{\partial J}{\partial \boldsymbol{\alpha}} - v^T \frac{\partial \boldsymbol{R}}{\partial \boldsymbol{\alpha}}, \tag{3}$$

The adjoint variable vector $v$ is determined by the adjoint equation:

$$\left(\frac{\partial \boldsymbol{R}}{\partial \boldsymbol{U}}\right)^T v = \left(\frac{\partial J}{\partial \boldsymbol{U}}\right)^T. \tag{4}$$

From Equation (3), assuming that $v$ and $\boldsymbol{U}$ are constant, the performance functional deviation caused by a geometric deviation can be obtained by the integral of:

$$\Delta J \approx \int_{\boldsymbol{\alpha}_0}^{\boldsymbol{\alpha}_0 + \Delta \boldsymbol{\alpha}} \frac{\partial J}{\partial \boldsymbol{\alpha}}(\boldsymbol{U}_0, \boldsymbol{\alpha}) \mathrm{d}\boldsymbol{\alpha} - v^T(\boldsymbol{U}_0, \boldsymbol{\alpha}_0) \int_{\boldsymbol{\alpha}_0}^{\boldsymbol{\alpha}_0 + \Delta \boldsymbol{\alpha}} \frac{\partial \boldsymbol{R}}{\partial \boldsymbol{\alpha}}(\boldsymbol{U}_0, \boldsymbol{\alpha}) \mathrm{d}\boldsymbol{\alpha}, \tag{5}$$

where, the subscript 0 represents the baseline, excluding the manufacturing deviation. As mentioned above, $U$ represents the flow solution, a 6×N vector. $U_0$ is the baseline flow solution vector. $\alpha_0$ represents the baseline geometry vector, $\Delta\alpha$ means the manufacturing deviation, which is the difference between the manufactured blade and the baseline geometry $\alpha_0$, the geometry of a manufactured blade with manufactured deviation can be expressed as $\alpha := \alpha_0 + \Delta\alpha$. Equation (5) can be expressed using the Newton–Leibniz formula:

$$
\begin{aligned}
\Delta J \approx & J(U_0, \alpha_0 + \Delta\alpha) - J(U_0, \alpha_0) \\
& - v^T(U_0, \alpha_0)[R(U_0, \alpha_0 + \Delta\alpha) - R(U_0, \alpha_0)].
\end{aligned}
\tag{6}
$$

The performance functional of the geometry defined by $\alpha_0 + \Delta\alpha$ is given by:

$$
\begin{aligned}
J(U, \alpha_0 + \Delta\alpha) := & \Delta J + J(U_0, \alpha_0) \\
\approx & J(U_0, \alpha_0 + \Delta\alpha) \\
& - v^T(U_0, \alpha_0)R(U_0, \alpha_0 + \Delta\alpha).
\end{aligned}
\tag{7}
$$

Referring to Equation (2), $R(U_0, \alpha_0)$ is dropped as it is zero. This formula defines the MC−adj−nonlinear method. It is capable of detecting the nonlinear correlation between the performance and the geometry.

By assuming that $dJ/d\alpha$ is constant in the interval $[\alpha_0, \alpha_0 + \Delta\alpha]$, Equation (5) can be simplified as:

$$
\Delta J \approx \left.\frac{dJ}{d\alpha}\right|_{U_0, \alpha_0} \Delta\alpha = \sum_{i=1}^{M} \frac{dJ}{d\alpha_i} \Delta\alpha_i.
\tag{8}
$$

where, $i \in [1, M]$ represents the $i^{th}$ geometric variable and the quantity of variables is denoted by M. The $i^{th}$ component of the vector $\Delta\alpha$ is $\Delta\alpha_i$. Now, the performance functional of the geometry defined by $\alpha_0 + \Delta\alpha$ can be approximated by:

$$
J(U, \alpha_0 + \Delta\alpha) \approx J(U_0, \alpha_0) + \sum_{i=1}^{M} \frac{dJ}{d\alpha_i} \Delta\alpha_i.
\tag{9}
$$

This is the MC−adj−linear method [19]. In comparison with the MC−adj−nonlinear method in Equation (7), the linear approach fails to take into account any nonlinearity of the performance functional related to the geometry variables.

It is conceivable that there exists a range for $\Delta\alpha$, in which Equation (7) is more precise than Equation (9). It is essential to note, however, that the range cannot be predetermined when used in a particular case. This work endeavors to assess the suitability of the MC−adj−nonlinear approach by comparing the UQ results of the MC−adj−nonlinear method to those of the MC−adj−linear method and the MC−CFD approach in two representative operational conditions with geometric variations originating from genuine manufactured blades.

In order to illustrate the MC−adj−linear method and the MC−adj−nonlinear method more clearly, the diagram of the procedures for the two methods are presented in Figures 1 and 2. In Figure 1, $\epsilon_i$ is a vector of the same size as $\alpha_0$. It represents a small perturbation to the $i$th element of the baseline geometric variable vector $\alpha_0$, thus, all but the $i$th element of $\epsilon_i$ are zero. $\Delta\alpha$ is a vector of the same size as $\alpha_0$. It represents the perturbation to $\alpha_0$ for a sample geometry.

As shown in Figure 1, a UQ analysis using the MC−adj−linear method involves three steps:

1. Obtain the flow solution $U_0$ and the adjoint solution $v_0^T$ for a specific performance functional for the baseline geometry;
2. Calculate sensitivity for all geometric variables. The sensitivity calculation involves mesh perturbation and the objective function $J$ & residual $R$ evaluation, based on the perturbed mesh $\alpha_0 + \epsilon_i$ and the baseline flow solution $U_0$. The finite difference method is also involved here to determine the sensitivities, see the green box in Figure 1. Therefore, the perturbation size of each design variable is quite important,

as too big or too small a value introduces big truncation or rounding errors. The operation has to be performed for each geometric variable (for 1:M, M represents the amount of the geometric variables). In this investigation, there were, in total, 398 geometric variables. This meant that the number of mesh perturbations and residual evaluations was 398;

3.  Calculate the performance metrics of all samples. The geometry perturbations of all samples($\Delta\alpha_i$) (for i = 1:N, N represents the amount of the samples) are reflected in changes to the baseline geometry variable vector $\alpha_0$. Then, Equation (9) is used to calculate the performance metric *J*.

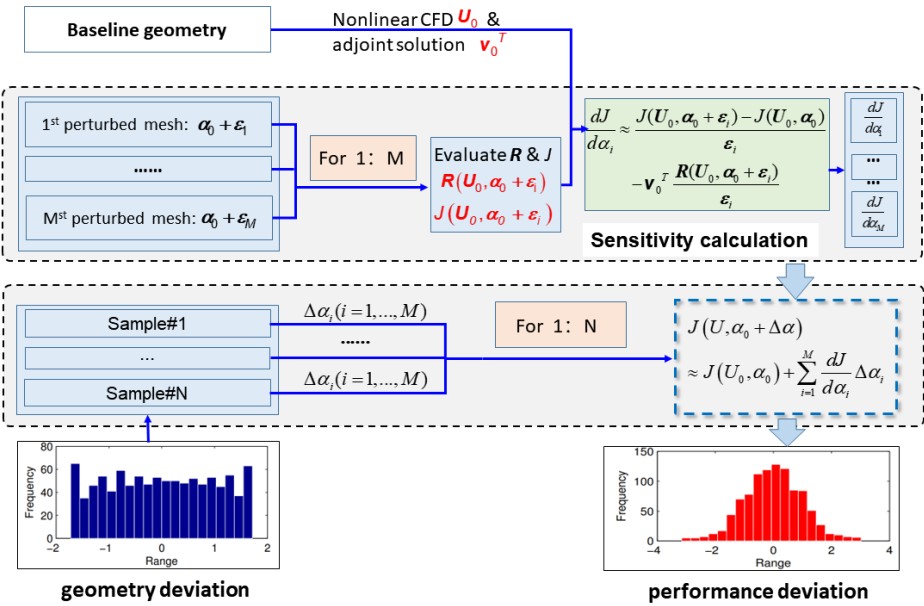

**Figure 1.** The diagram of the MC−adj−linear method procedure.

Figure 2 illustrates the MC−adj−nonlinear method procedure. It involves two steps in an UQ analysis. The initial stage is identical to that of the MC−adj−linear method. In the subsequent step, the sample's performance metric is determined using Equation (7), by perturbing the baseline geometry $\alpha_0$ by adding $\Delta\alpha$ (that is the geometry of the sample, totally N times). In contrast to the MC−adj−linear technique, this nonlinear approach does not rely on finite difference, thus, there are no truncation or rounding errors. In an UQ analysis using this method, the number of mesh perturbations and residual evaluations is equal to the number of samples. If the amount of samples greatly exceeds the amount of geometry variables, then the MC−adj−nonlinear method is more time-consuming than the MC−adj−linear method. Actually, we cannot draw a general conclusion as to which method is cheaper and more recommended, as the conclusion lies in the case. The amount of time it takes to use the MC−adj−linear approach and the MC−adj−nonlinear approach is dependent upon the amount of geometric variables and samples. If the amount of geometry variables, M, is more than the amount of samples, N, of the case, then the MC−adj−nonlinear is more recommended. Otherwise, the MC−adj−linear is more recommended in terms of the time cost. However, we can confirm that no matter whether the MC−adj−linear approach or the MC−adj−nonlinear approach is used, the time cost for both these methods is at least ten times less than what is required by the MC−CFD method.

The use of both the MC−adj−linear method and the MC−adj−nonlinear method requires a nonlinear flow solver and an adjoint solver. In this work, the in-house flow solver, named Newton Unstructured Turbomachinery Solver for Computational Fluid Dynamics (NutsCFD), was employed. A Jacobian forming Newton–Krylov method, Generalized Minimal RESidual (GMRES) method, was employed to solve the large sparse linear system

of the equation. A discrete adjoint solver was also developed, on the basis of the flow solver NutsCFD. As the system of the Jacobian matrix is explicitly formed for solving the nonlinear flow equation, the Jacobian matrix was transposed to construct an adjoint system. The solution of the adjoint system was also accomplished using the GMRES method. As the adjoint equation system is a linear system, a single GMRES solution is required. Therefore, the time cost of solving the adjoint equation using the GMRES method is just a fraction of that of solving the nonlinear flow equation, which uses the GMRES solution as part of a nonlinear iteration process. The small time cost of an adjoint solution, with respect to that of a nonlinear solution, differs from that resulting from a solution method using a fixed point iteration. Further information regarding the flow/adjoint solver can be obtained from ref. [24–26].

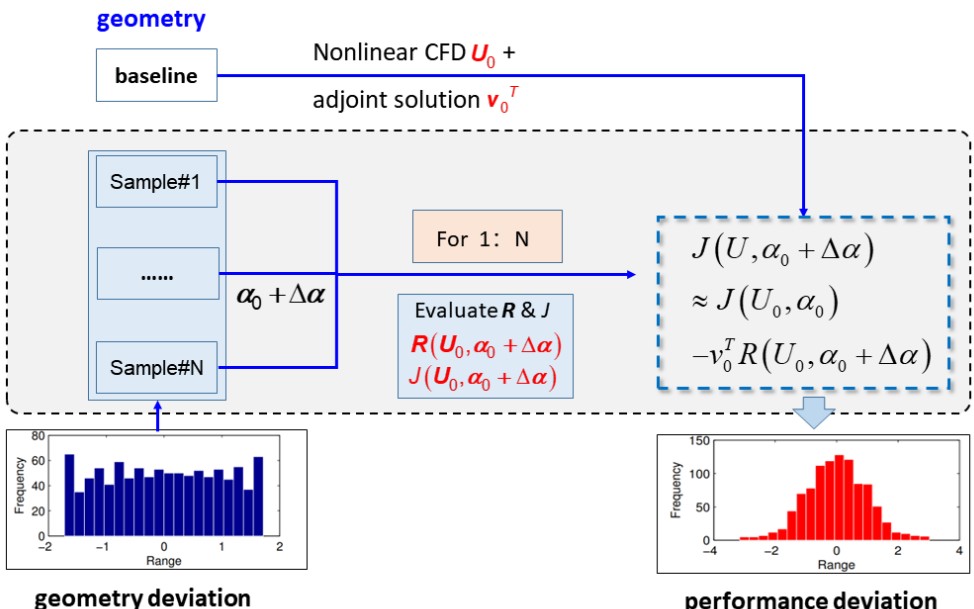

**Figure 2.** The diagram of the MC−adj−nonlinear method procedure.

## 3. Uncertainty Quantification

This section concentrates on the UQ of the performance variations resulting from manufacturing deviations in a high compressor blade section. The suitability of the MC−adj−nonlinear method and its superiority to the other two methods, MC−CFD and MC−adj−linear, were studied at two specific conditions, through the utilization of the three methods.

### 3.1. Test Case

To minimize the computational expense of the MC−CFD approach, which served as the benchmark, a transonic compressor blade section, located at a radius of 0.245 m, was selected. The compressor's design speed was 11,990 RPM. In this work, the geometric variables were the coordinate points at the blade surface. The profile of the sectional blade section was outlined by the coordinates of 398 points on its surface. The areas around the leading edge and trailing edge (LE and TE) of the blade had higher concentrations of surface points, as illustrated in Figure 3.

The computational mesh was generated using the NUMECA Autogrid software and consisted of a total number of 6522 quadrilateral elements, as shown in Figure 4. At the inlet of all CFD analyses domains, the total pressure and temperature were set to 158,885.9 Pa and 330.6 K, respectively, with an axial incoming flow. A constant pressure was used as the outlet condition. The flow solution (relative Mach number distribution) at the design point is shown in Figure 5.

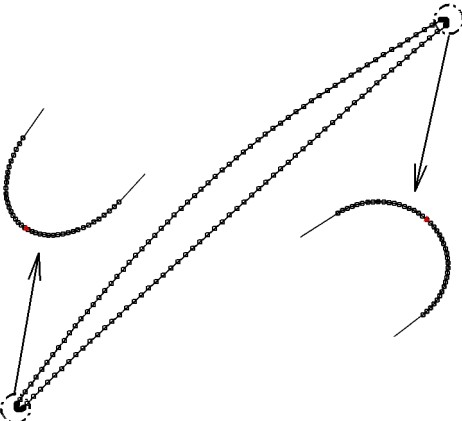

**Figure 3.** Blade profile and distribution of control points.

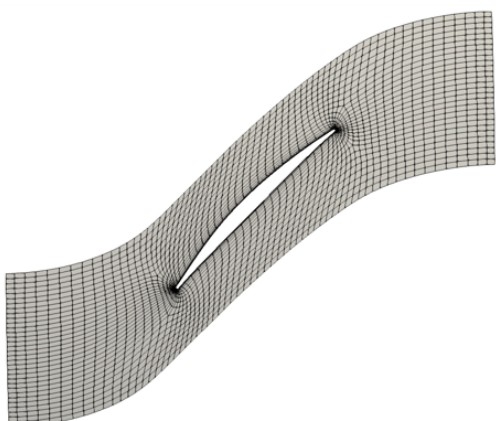

**Figure 4.** The mesh used in this study.

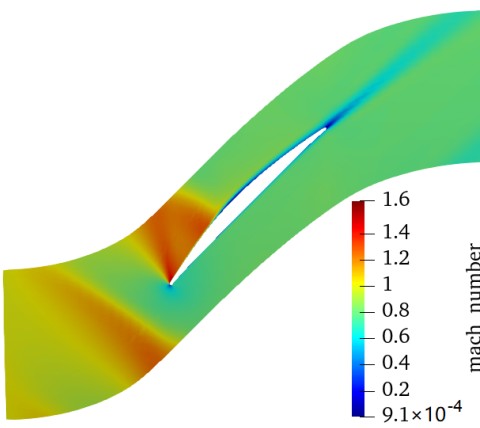

**Figure 5.** The relative Mach number distribution at the design point.

To obtain the performance UQ due to geometric variations, geometric perturbations in the normal direction to the 398 points were added to the baseline geometry, bringing the total number of samples to 2899. The added geometric perturbations had a standard deviation and a mean (The actual values are not provided at the request of the data owner) derived from the scanned data of manufactured blades. To reflect the correlation between perturbations to the coordinates of neighboring points, the coordinate perturbations of some points on the blade surface were generated first, and the coordinate perturbations to the points in between were interpolated using a smooth function. The nominal blade, average blade, and the average blades with twice the positive and negative standard deviation

are shown in Figure 6, where, $\sigma$ represents the standard deviation of the manufactured deviation.

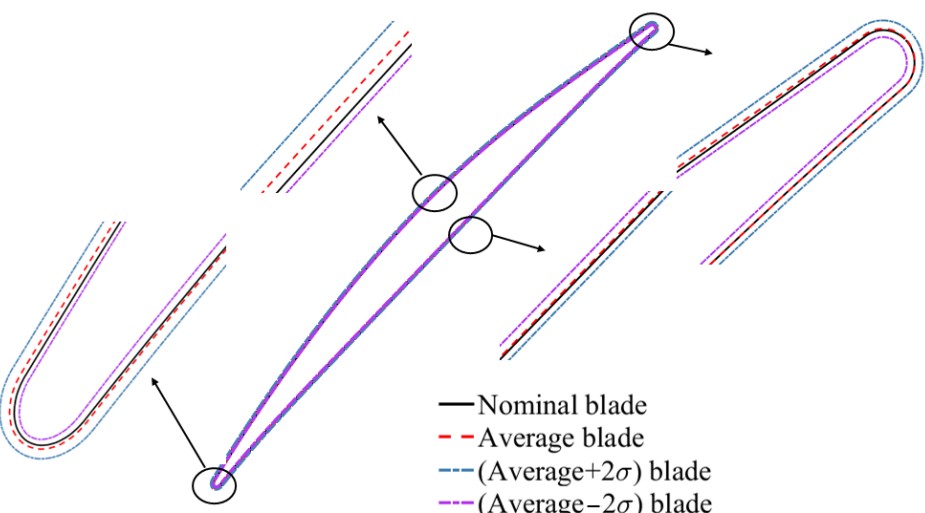

**Figure 6.** The nominal blade, average blade, and the average blades with twice the positive and negative standard deviation.

*3.2. Adjoint Solution Verification*

Both MC−adj−linear and MC−adj−nonlinear methods need the adjoint solution. Hence, in order to evaluate the reliability of the adjoint solution, this subsection compares the sensitivity of a single-mode profile perturbation, which is expressed by $\epsilon$, between the finite difference and the adjoint methods. The adjoint solution of the eddy viscosity with the mass flow rate as the performance functional at the design point is shown in Figure 7.

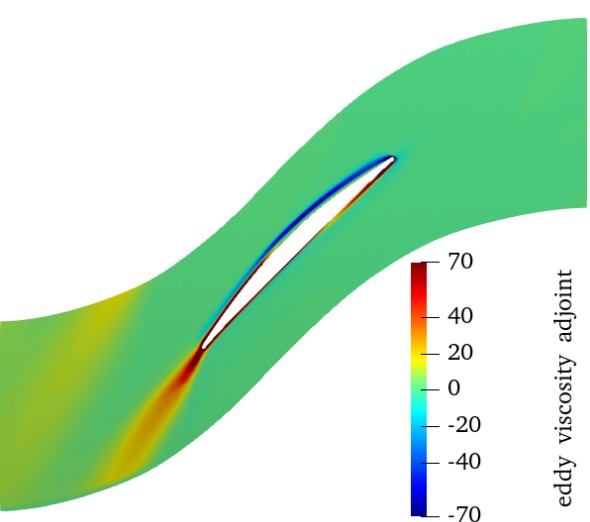

**Figure 7.** The eddy viscosity adjoint field with the mass flow rate as the performance functional at the design point.

Figure 8 displays the sensitivities of mass flow rate ($\dot{m}$), pressure ratio ($\pi$), and efficiency ($\eta$) computed using the two methods when the perturbation step size is altered within the range between $10^{-8}$ and 1 times $\epsilon$. As can be seen in Figure 8, when the step size was between $10^{-6}$ and $10^{-3}$, the sensitivities of the three quantities were almost constant. The relative difference between the adjoint sensitivity and the finite difference sensitivity was below 0.01% when the step size was within the range of $[10^{-5}, 10^{-4}]$. The precision

of the adjoint sensitivity was acceptable. It should be noted that the adjoint sensitivity calculation also involved finite differencing to calculate residual perturbations due to mesh perturbations. The use of finite differencing led to the variation of the calculated adjoint sensitivities with respect to the perturbation size.

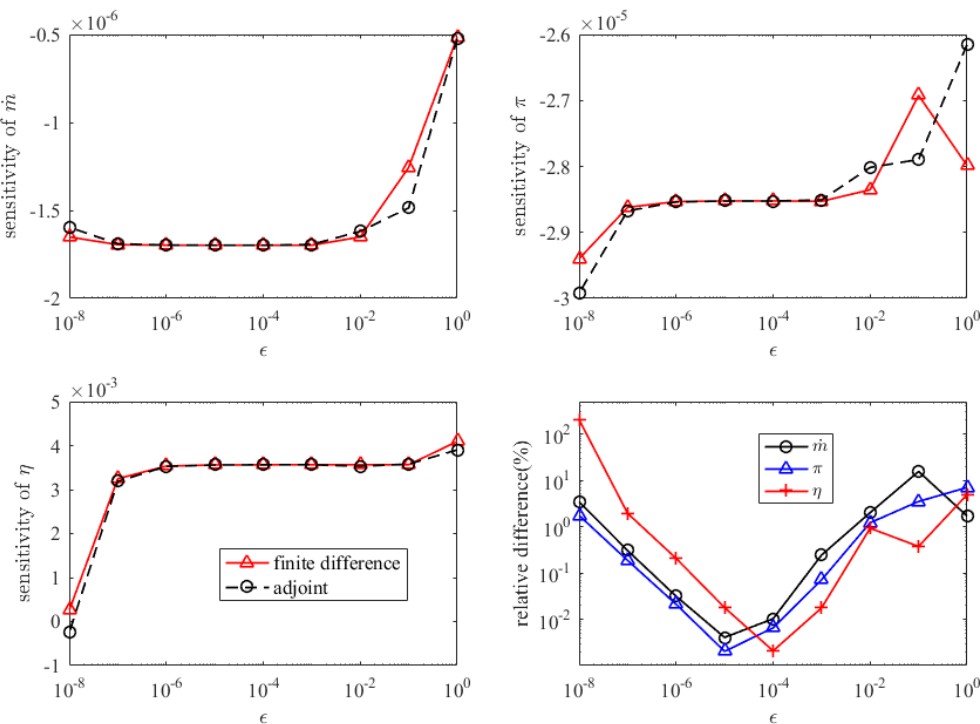

**Figure 8.** Comparison of sensitivities between the finite difference and the adjoint methods.

### 3.3. Verification of the MC−Adj−Nonlinear Method at Design and Near-Stall Conditions

Figures 9 and 10 display the performance map of the nominal blade, the aerodynamic performance scatterings and the probability density functions (PDFs) of the samples at the design and the near-stall points computed utilizing the three methods. The different colors of all the dashed lines, curves, and dots in Figures 9 and 10 represent the results calculated by different methods: black corresponds to the MC−CFD method, red corresponds to the MC−adj−linear method, and blue corresponds to the MC−adj−nonlinear method. The dashed lines represent the mean of the mass flow rate, pressure ratio, and efficiency at the two conditions calculated by the three methods. The curves without dots represent the PDFs of the mass flow rate, pressure ratio, and efficiency computed by the three methods. The dots are the objective performance of all samples at the two conditions calculated by the three methods. The black curve with dots is the performance map of the baseline calculated by CFD simulation. There appeared to be a definite connection between pressure ratio/efficiency and mass flow rate: when the mass flow rate increased, the pressure ratio and efficiency also increased. The PDFs calculated by the MC−adj−nonlinear method showed greater consistency with those of the MC−CFD method, whereas the PDFs calculated by the MC−adj−linear method exhibited a tendency towards higher values with more prominent peaks than the MC−CFD method. Table 1 presents the mean mass flow rate, pressure ratio, and efficiency at the design and near-stall points, computed using the three methods( the mean is also shown by the dotted lines in Figures 9 and 10).

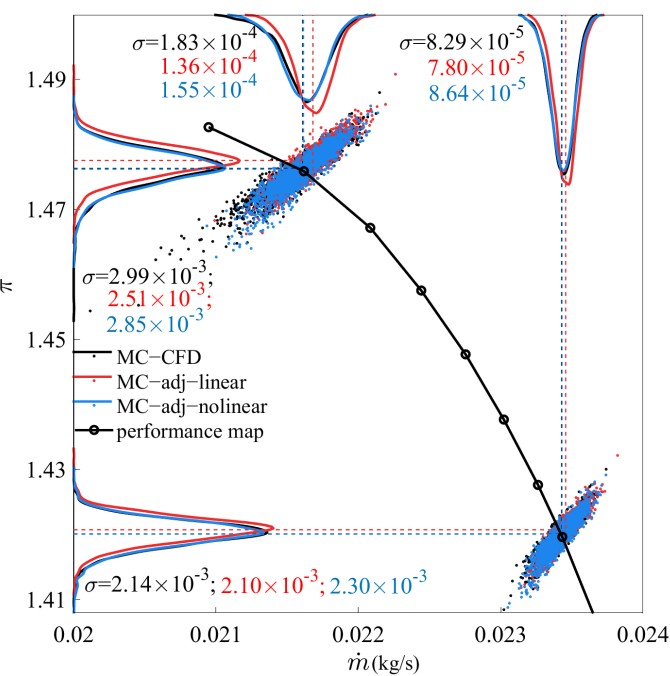

**Figure 9.** Performance characteristic with PDFs of mass flow rate and pressure ratio using the three methods.

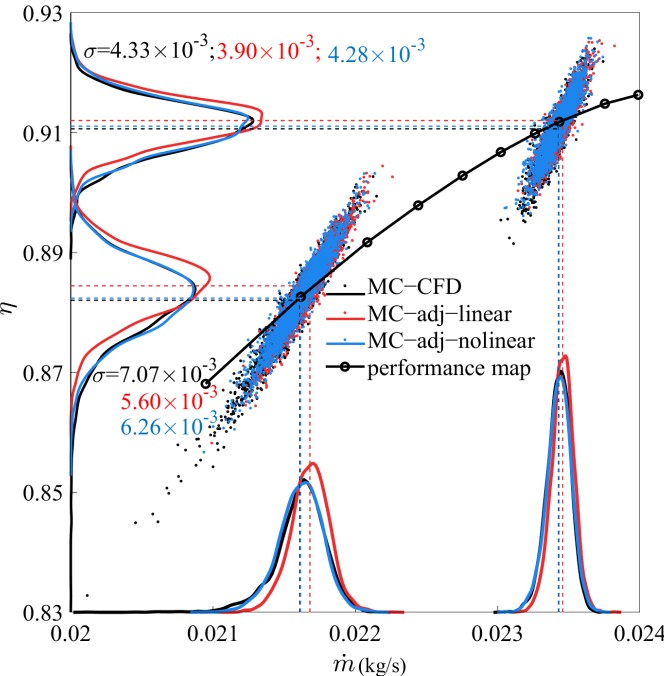

**Figure 10.** Performance characteristic with PDFs of mass flow rate and efficiency using the three methods.

The absolute difference in a performance functional, be it mass flow rate, pressure ratio, or efficiency, between the three methods, was very small. To contrast the differences, the mean deviations of the three quantities, normalized to their respective baseline values, are shown in Figure 11. In the figure, $J$ represents $\dot{m}$, $\pi$, or $\eta$, the subscript $_0$ represents the baseline and $\Delta$ denotes the difference in a performance functional between a sample and the baseline.

**Table 1.** The mean aerodynamic performance computed using the three methods at the design and near-stall points.

| Operating Point | Numerical Method | Mass Flow Rate (kg/s) | Pressure Ratio | Efficiency |
|---|---|---|---|---|
| the design condition | MC−CFD | 0.023428 | 1.420063 | 0.910661 |
| | MC−adj−linear | 0.023457 | 1.420703 | 0.912055 |
| | MC−adj−nonlinear | 0.023431 | 1.420036 | 0.911103 |
| the near-stall condition | MC−CFD | 0.021610 | 1.476226 | 0.882060 |
| | MC−adj−linear | 0.021681 | 1.477472 | 0.884464 |
| | MC−adj−nonlinear | 0.021614 | 1.476129 | 0.882412 |

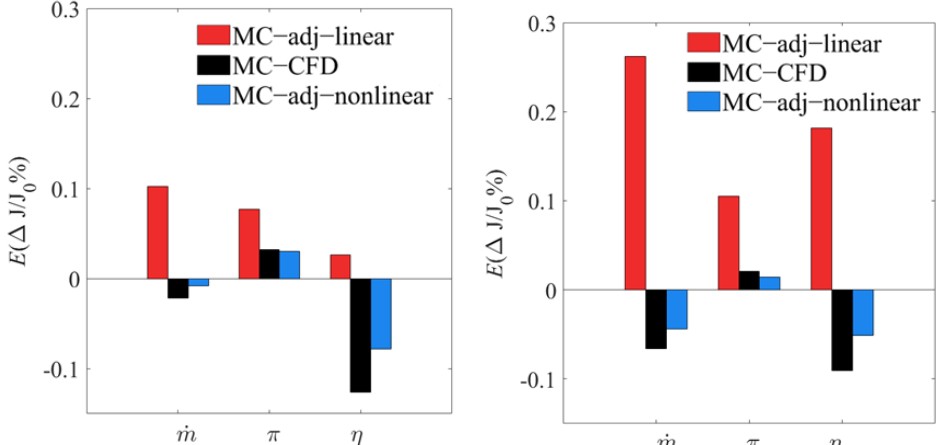

**Figure 11.** The mean of normalized performance variations at the design (**left**) and near-stall conditions (**right**) computed using the three methods.

It can be seen that the sign of the mean of normalized performance deviations predicted by the MC−adj−linear were wrong. Although there were some discrepancies in the mean of normalized performance deviations between the MC−CFD method and the MC−adj−nonlinear method, the sign of the mean by the MC−adj−nonlinear method was always correct for this particular case. This meant that the MC−adj−nonlinear method was much more reliable than the MC−adj−linear method. This feature is also critical for robust optimization when using the adjoint approach.

The calculated standard deviations of mass flow rate, pressure ratio, and efficiency at the two operating conditions are presented in Figures 9 and 10. Figure 12 compares the normalized standard deviations of the three quantities. The standard deviations were normalized by their respective baseline performance functionals. The difference in the normalized standard deviations, using the three methods, was much smaller than in the mean. At the design condition, the results from the three methods were very close. Nevertheless, it can be seen that the predictions from the MC−adj−nonlinear method were closer to those from the MC−CFD method than those from the MC−adj−linear method. At the near-stall condition, the difference in the standard deviation was much more pronounced among the three methods. Both the MC−adj−linear method and the MC−adj−nonlinear method under-predicted the values. Nevertheless, the predictions from the MC−adj−nonlinear were once again closer to those of the MC−CFD method than those of the MC−adj−linear method.

Figures 11 and 12 show that the MC−adj−nonlinear method was more accurate than the MC−adj−linear method. This was because the MC−adj−nonlinear method could capture some nonlinear effects and there was a non-negligible nonlinear effect in the performance functionals at the design speed. The nonlinear effect was expected to be

pronounced at the near-stall condition, where a larger difference was observed between the results calculated by the linear and nonlinear methods. It was expected that the nonlinearity of the flow at a low speed would be weaker and the two methods were expected to give smaller differences. To verify this speculation, the mean and standard deviation of normalized aerodynamic performance deviations at two operating points with different pressure ratios at 50% speed were calculated by the two adjoint-based methods. Figures 13 and 14 show the mean and the standard deviation of normalized performance deviations of the two operating points at 50% speed, together with the results of 100% speed. We found that the difference in the UQ results between the MC−adj−linear method and the MC−adj−nonlinear was much smaller at 50% speed than at the 100% speed, as expected. In particular, at 50% speed, the mean of the performance deviations predicted by the two methods had the same correct signs. This further illustrated that the MC−adj−nonlinear method could capture some nonlinear effects.

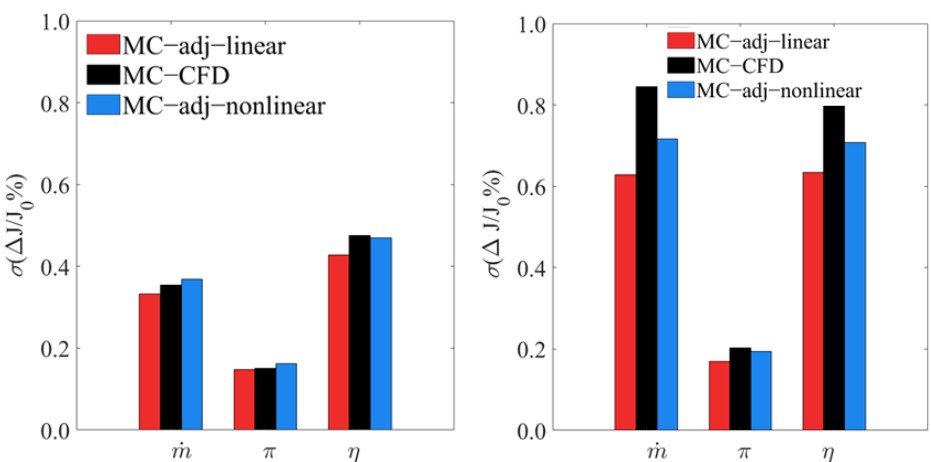

**Figure 12.** The normalized standard deviation of performance functionals variation at the design (**left**) and near-stall conditions (**right**) computing using the three methods.

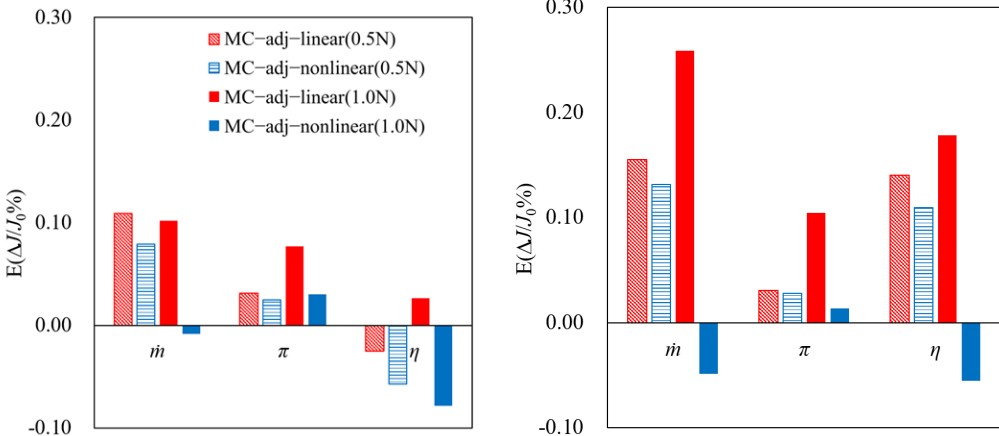

**Figure 13.** The mean of normalized performance deviations at the design (**left**) and near-stall conditions (**right**) of 100% speeds and that of low pressure ratio (**left**) and high pressure ratio (**right**) operating points at 50% speed by the MC−adj−linear and MC−adj−nonlinear methods.

Table 2 presents the time-consumption of performing the UQ for this case at the design condition using the three methods. The MC−CFD method solved the nonlinear flow equation 2899 times which accounted for 97% of the total time cost. Both the MC−adj−linear method and the MC−adj−nonlinear method solved the nonlinear flow equation once (for the baseline geometry only) and the adjoint equation three times (for the baseline geometry

and three performance functionals). With the use of the GMRES method to solve a linear system of equation, the time cost of an adjoint solution was about 6.7% that of a nonlinear flow solution. For both the MC−adj−linear method and the MC−adj−nonlinear method in this case, the major time cost came from the mesh perturbation, which accounted for about 84% and 90% of the total, respectively. It should be noted that the relative time cost of each component in Table 2 is valid for two-dimensional cases only and is not valid for three-dimensional cases. For a three-dimensional case with grid points in the order of 0.5 million, the relative time cost of solving the nonlinear flow equation and the adjoint equation would increase considerably in comparison with that of grid perturbation and residual evaluation.

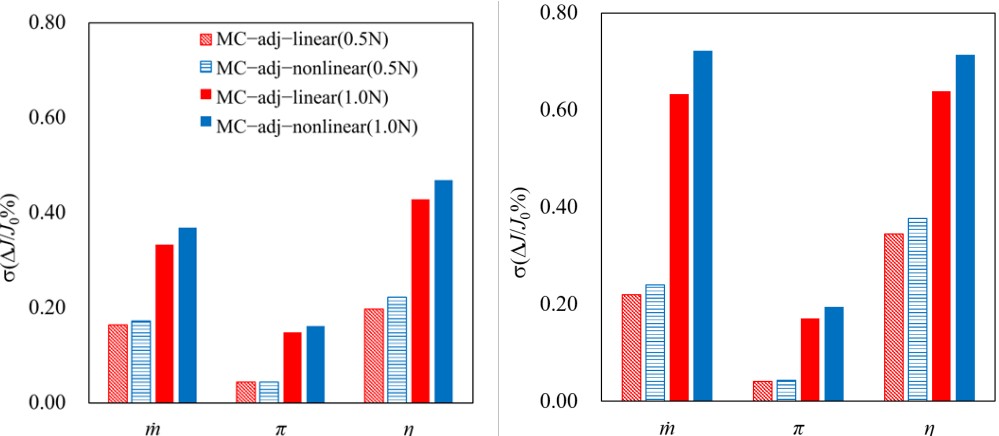

**Figure 14.** The normalized standard deviation of performance functionals at the design (**left**) and near-stall conditions (**right**) at 100% speeds and that of low pressure ratio (**left**) and high pressure ratio (**right**) operating points at 50% speed by the MC−adj−linear and MC−adj−nonlinear methods.

**Table 2.** The UQ time-consumption of the three methods at the design condition (eight cores).

| Time (min) | Grid | Grid of Baseline | Cfd | Adjoint | Residual | Sum | Time Ratio |
|---|---|---|---|---|---|---|---|
| MC−CFD | $0.017 \times 2899$ | 0 | $0.539 \times 2899$ | 0 | 0 | 1611.84 | 203.68 |
| MC−adj−linear | $0.017 \times 389$ | 0.017 | 0.539 | $0.036 \times 3$ | $0.0016 \times 398$ (for Equation (8)) | 7.91 | 1 |
| MC−adj−nonlinear | $0.017 \times 2899$ | 0.017 | 0.539 | $0.036 \times 3$ | $0.0016 \times 2899$ (for Equation (6)) | 54.58 | 6.90 |

The total time cost ratio of the MC−CFD, the MC−adj−linear, and the MC−adj−nonlinear method was about 204:1:7. The MC−adj−nonlinear method was roughly seven times more time-consuming than the MC−adj−linear, due to the fact that it required 2899 mesh perturbations and residual evaluations, while the MC−adj−linear only needed 389 mesh perturbations and residual evaluations, a number that is roughly seven times the number of geometric variables in this study.

It was demonstrated that the MC−adj−nonlinear technique provided an appropriate balance between accuracy and time-consumption. The mean of the mass flow rate at the near-stall condition was selected to quantify the accuracy and time cost of the three methods. In Figure 15.

The MC−CFD method yielded an error of zero because its results were used as the benchmark, although it was the most time-consuming approach. The MC−adj−linear method required the least amount of time but resulted in the biggest error. The time-consumption of the MC−adj−nonlinear method was slightly bigger than that of the MC−adj−linear method but the solution error was greatly reduced, compared with that of the MC−adj−linear method. Hence, it was evident that the MC−adj−nonlinear approach offered a good balance of precision and time consumption for the UQ.

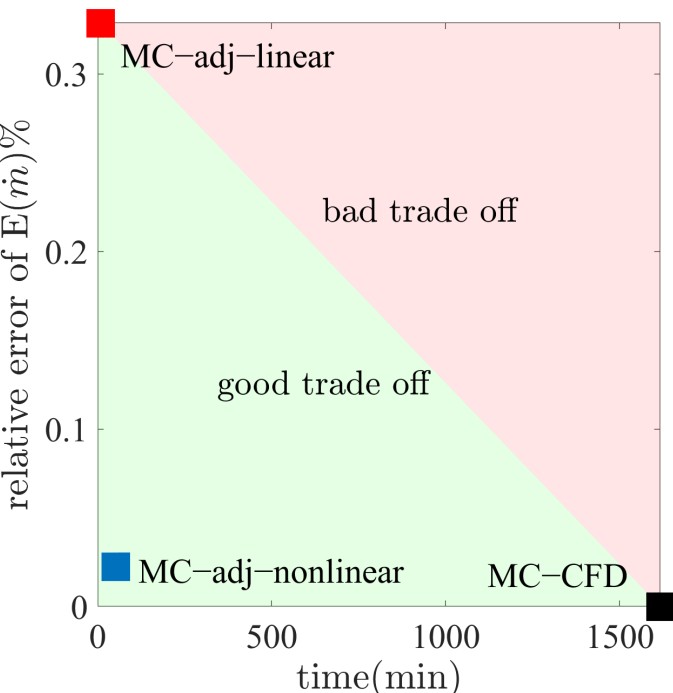

**Figure 15.** Time cost versus accuracy in calculating the mean mass flow rate at the design condition using the three methods.

*3.4. Full Map UQ of Aerodynamic Performance at Four Speeds*

With the applicability of the MC−adj−nonlinear method for UQ demonstrated above, the method was used to investigate how the geometric deviations affected the aerodynamic performance of the blade section over a wider operating range. To do this, the UQs of mass flow rate, pressure ratio, and efficiency were performed at four speeds: 50%, 80%, 100%, and 120% of nominal speeds.

Figure 16 shows the pressure ratio and efficiency characteristics of the baseline design and all 2899 samples. The characteristics of the baseline design were obtained using CFD, while those of the 2899 samples were obtained using the MC−adj−nonlinear method. The green curves in Figure 16 represent the characteristics of sample #2839 (marked as sample_W with W standing for worse in Figure 16), the performance of which was always worse than that of the baseline. The purple curves correspond to sample #1159 (marked as sample_B with B standing for better in Figure 16), the performance of which was always better than that of the baseline. In order to understand the cause of the performance deviations of sample_B and sample_W from the baseline, Figure 17 shows the profiles of the baseline, sample_W, and sample_B. It can be seen that the main differences between the three blades were in the leading edge and the suction side. Sample_B had a thinner leading edge and an increased blade inlet angle. Sample_W had a thicker leading edge and a reduced blade inlet angle.

To facilitate further understanding of the performance deviations of sample_B and sample_W from the baseline, the relative Mach number contours of the two blades and the baseline blade at the near-stall conditions of the four speeds are presented in Figures 18–21. It can be seen that at the near-stall conditions, flow was separated on the blade suction surface. Sample_B had the smallest separation bubble, while sample_W had the biggest separation bubble. The difference in the separation bubble size was attributed to the change in the blade inlet angle and the blade leading edge thickness. Both the reduction of the blade leading edge thickness and the increase of the blade inlet angle were favorable in reducing the flow separation. At 120% speeds, the strength of the detached shock was reduced and pushed further upstream for sample_B (marked with the red box in Figure 21).

The former could be attributed to the reduction of the blade leading edge thickness and the latter was attributed to the increase of the blade inlet angle.

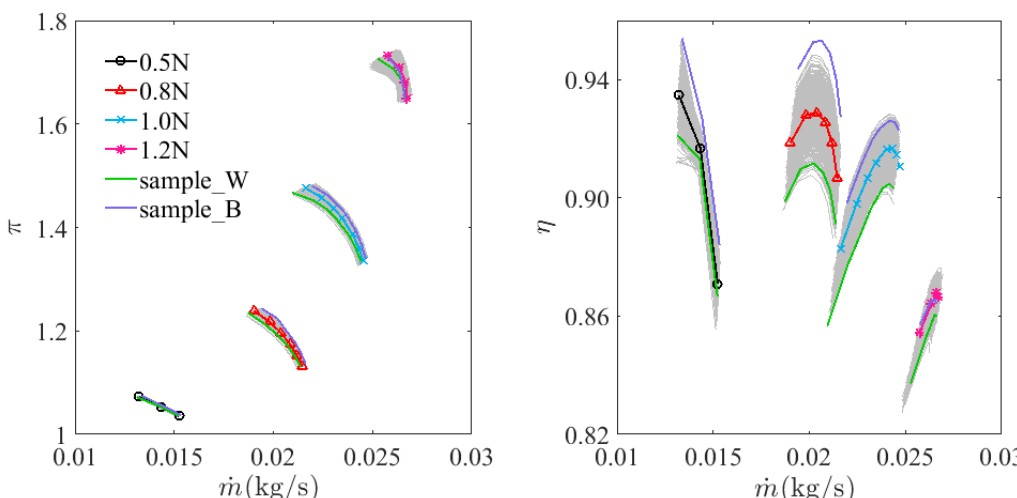

**Figure 16.** Pressure ratio and efficiency characteristics at different speeds.

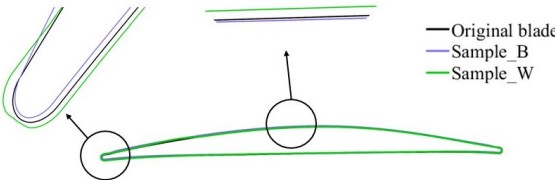

**Figure 17.** The profiles of the baseline, sample_W, and sample_B.

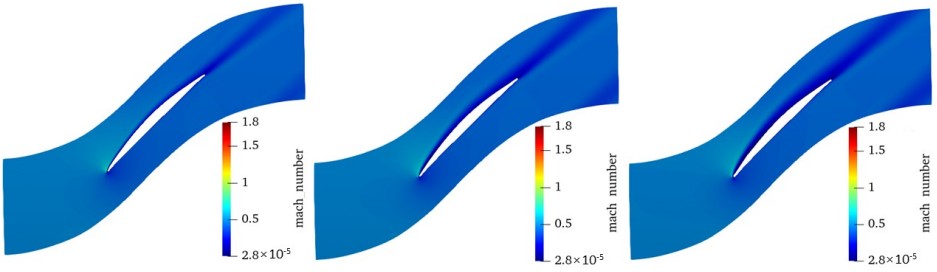

**Figure 18.** Mach number distribution of the three blades at the near-stall condition at 50% speed: sample_B (**left**), baseline (**middle**) and sample_W (**right**).

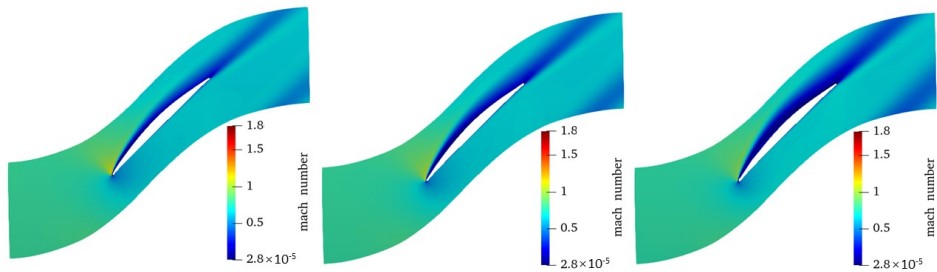

**Figure 19.** Mach number distribution of the three blades at the near-stall condition at 80% speed: sample_B (**left**), baseline (**middle**) and sample_W (**right**).

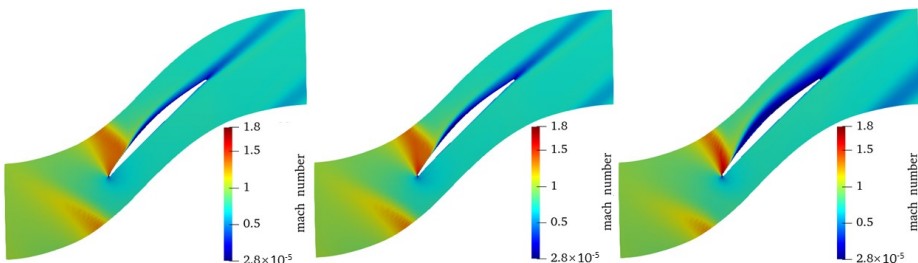

**Figure 20.** Mach number distribution of the three blades at the near-stall condition at 100% speed: sample_B (**left**), baseline (**middle**) and sample_W (**right**).

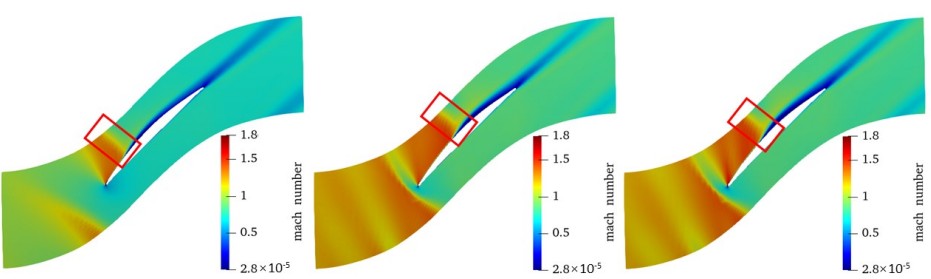

**Figure 21.** Mach number distribution of the three blades at the near-stall condition at 120% speed: sample_B (**left**), baseline (**middle**) and sample_W (**right**).

The mean and standard deviation of mass flow rate, pressure ratio, and efficiency are presented in Figures 22–24.

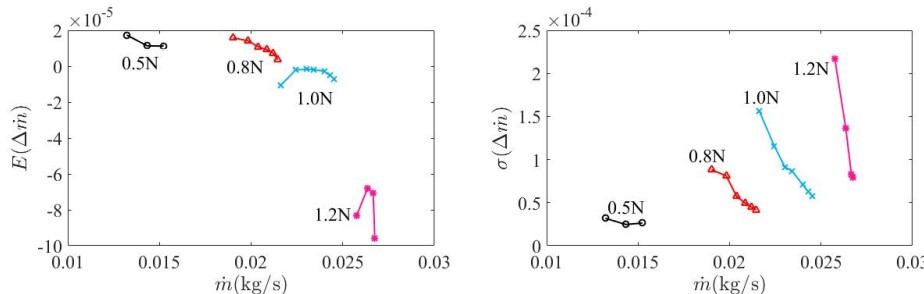

**Figure 22.** Uncertainty statistics of mass flow rate.

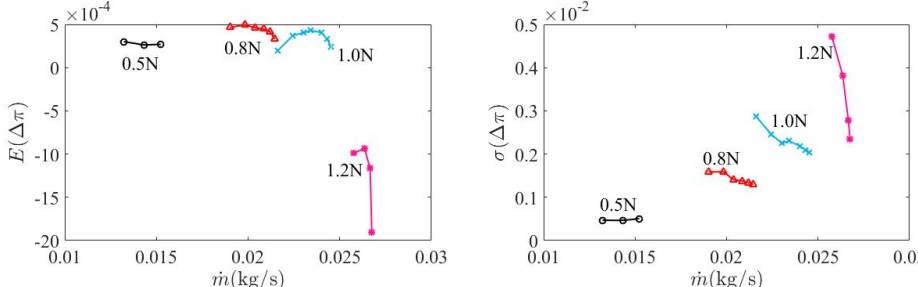

**Figure 23.** Uncertainty statistics of pressure ratio.

It should be noted that the mean is shifted by the baseline value, which can be considered the mean of deviation. Presented in this way, the sign of the mean indicates whether the mean of performance functional increased or reduced with respect to the baseline value. It should also be noted that the standard deviation of a shifted performance

functional remains the same as its original unshifted quantity, as the standard deviation represents the scattering of a quantity and it should not change from data shifting.

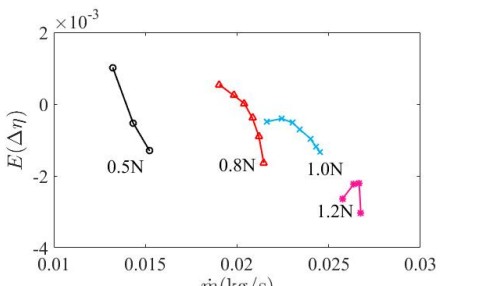 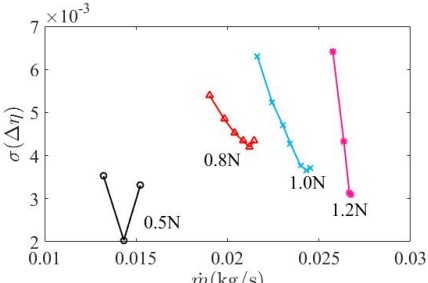

**Figure 24.** Uncertainty statistics of efficiency.

From Figure 22, we can see that the effect of geometric deviations on the mean mass flow rate was different at low and high speeds. At low speeds (50% and 80% speeds), the mean mass flow rate increased. At high speeds(100% and 120% speeds), the mean mass flow rate reduced. Even at the two low speeds, the increase of the mean mass flow rate (the mean mass flow rate deviation) reduced from the stall side to the choke side. The mean mass flow rate variation could be attributed to the increased blade inlet angle and the increased thickness of the average blade (the major thickness increase was on the blade suction side, see Figure 6). An increased blade inlet angle was beneficial for operating conditions with a large inlet flow angle (the two low speeds). With higher inlet flow velocity, the mass flow rate and other performance functionals were more sensitive to geometric blockage, due to the thickening of the blade suction side.

At 50% speed, the insensitivity of mass flow rate to geometric deviations led to a small standard deviation. The standard deviation of mass flow rate increased with the increase of the shaft speed. Along a speedline, the standard deviation increased from the choke side to the stall side. The increase in standard deviation became more and more pronounced with increase in the shaft speed.

The effect of geometric deviations on uncertainty statistics of pressure ratio at the four speeds was qualitatively the same as that of mass flow rate. However, the effect of geometric deviations on uncertainty statistics of efficiency was different from that of mass flow rate. The geometric deviations were largely detrimental to efficiency. Only at the near stall region of 50% and 80% speeds was the mean efficiency increased. In general, the mean efficiency deviation increased with the increase of the shaft speed. This implied that geometric deviations are more detrimental to efficiency at high speeds. Regarding the standard deviation, it increased along a speedline from the choke side to the stall side, which was the same as that of mass flow rate and pressure ratio. Across the four speeds, at both the choke and stall sides, the standard deviation increased first and then reduced.

## 4. Conclusions

The paper researched the applicability of the adjoint-based nonlinear method for fast UQ of aerodynamic performance of compressor blades and the effects of real-life manufacturing deviations on performance deviations using the adjoint-based nonlinear method. The following three conclusions are determined:

(1) At 100% speed, compared with the MC−adj−linear method, the UQ results predicted by the MC−adj−nonlinear method were more accurate, especially for the near-stall condition, where the nonlinear dependence of performance functionals on geometric variables was stronger. At 50% speed, the differences in the UQ results predicted by the two adjoint-based methods were much smaller, due to the weaker nonlinearity of the flow. The MC−adj−nonlinear method required nearly 30 times less time than the MC−CFD method. Hence, the MC−adj−nonlinear approach provides a satisfactory balance between precision and time cost for UQ.

(2) Aerodynamic performance is more sensitive to geometric deviations at high speeds than at low speeds. For this particular case, the geometric deviations produced an increased mean of mass flow rate and pressure at low speeds, while incurring a reduced mean at high speeds. The geometric deviations were generally detrimental to the mean efficiency over the four speeds. The reduction of the mean of mass flow rate, pressure ratio, and efficiency became more with increase in shaft speed.

(3) The standard deviation of performance generally increased with increase in shaft speed. Along a speedline, the standard deviation also increased with increase in pressure ratio. The difference in standard deviation between a near choke point and a near stall point along a speedline was much larger at high speeds than at low speeds.

Further studies will be conducted to examine the effectiveness of the approach on three-dimensional turbomachinery components.

**Author Contributions:** Conceptualization, S.X. and Q.Z.; methodology, S.X. and Q.Z.; software, S.X.; validation, Q.Z.; formal analysis, Q.Z., S.X. and D.W.; investigation, Q.Z.; resources, S.X. and Q.Z.; data curation, Q.Z., D.W. and S.X.; writing—original draft preparation, Q.Z.; writing—review and editing, D.W. and S.X.; visualization, Q.Z.; supervision, S.X., D.W. and X.H.; project administration, S.X. and Q.Z.; funding acquisition, S.X., D.W. and X.H. All authors have read and agreed to the published version of the manuscript.

**Funding:** This research was funded by the National Natural Science Foundation of China (Grant No.52006177) and National Science and Technology Major Project (Grant No.2017-II-0009-0023).

**Data Availability Statement:** The data are not publicly available due to the row geometric deviation, which is the base of this study, is not allowed to be public.

**Acknowledgments:** The authors gratefully acknowledge the data support from Xianjun Yu and Jiaxin Liu at Beihang University

**Conflicts of Interest:** The authors declare no conflict of interest.

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
