# Peer review of "Uncertainty Quantification of Compressor Map Using the Monte Carlo Approach Accelerated by an Adjoint-Based Nonlinear Method"

_aerospace, doi:10.3390/aerospace10030280_

Round 1

Reviewer 1 Report

The presented work compares different methods (adjoint-based linear, adjoint-based nonlinear and CFD) of uncertainty quantification of compressor performance. The adjoint-based nonlinear method is identified as effective method having low resulting errors, while being efficient in computational cost.

The topic is of interest to compressor developers, manufacturers and users since the developed method promises either an increase in precision or a decrease in computational time and cost. Compressor users will gain more insight into the development process of compressors and will have a better understanding of uncertainties in compressor maps. Compressors of any type and shape are in general of interest in the aviation sector, therefore the presented work fits well into the scope of Aerospace.

The text is in general very well written, the reader is able to follow the authors thoughts. The results are explained clearly. Both methodology and results are good.

To further improve the paper:

·       Chapters 1 & 2 could be extended to better introduce the methods, in order to widen the audience by making the topic more accessible to readers who are not yet familiar with it.

·       Figures 1,  2 , 7 and 8 should be improved and better explained in the text, otherwise the reader is not able to understand the information in them.

·       Variables like epsilon, pi, sigma should be explained in the text. The axis description in figure 6 is too small.

·       Figure 8 needs a better explanation what the lines and dots are showing.

·       Line 218: of nominal or maximum speed?

·       The cited literature is relevant, although more literature could be included.

n  Some more minor points can be found as comments in the pdf

Author Response

the response are in the WORD file, thanks for the nice suggestions

Reviewer 2 Report

Dear authors,

please refer to the PDF file with comments.

Bets regards.

Author Response

(The authors gave the same response as above.)

Round 2

Reviewer 2 Report

Dear author,

thank you for the extensive revision of the manuscript following my recommendations.

I feel the manuscript is worth publishing in its current form.

Best regards.